# Cellulose Acetate Phthalate-Based pH-Responsive Cyclosporine A-Loaded Contact Lens for the Treatment of Dry Eye

**DOI:** 10.3390/ijms24032361

**Published:** 2023-01-25

**Authors:** Jonghwa Kim, Himangsu Mondal, Rujun Jin, Hyeon Jeong Yoon, Ho-Joong Kim, Jun-Pil Jee, Kyung Chul Yoon

**Affiliations:** 1Department of Ophthalmology, Chonnam National University Medical School and Hospital, Gwangju 61469, Republic of Korea; 2College of Pharmacy, Chosun University, Gwangju 61452, Republic of Korea; 3Department of Chemistry, Chosun University, Gwangju 61452, Republic of Korea

**Keywords:** dry eye, cyclosporine A, contact lens, supercritical fluid technique

## Abstract

Cyclosporine A (CsA) as an eye drop is an effective treatment for dry eye. However, it has potential side effects and a short ocular residence time. To overcome these obstacles, we developed a cellulose acetate phthalate-based pH-responsive contact lens (CL) loaded with CsA (CsA-CL). The CsA was continuously released from the CsA-CL at physiological conditions (37 °C, pH 7.4) without an initial burst. CsA was well-contained in the selected storage condition (4 °C, pH 5.4) for as long as 90 days. In safety assays, cytotoxicity, ocular irritation, visible light transmittance, and oxygen permeability were in a normal range. CsA concentrations in the conjunctiva, cornea, and lens increased over time until 12 h. When comparing the therapeutic efficacy between the normal control, experimental dry eye (EDE), and treatment groups (CsA eye drop, naïve CL, and CsA-CL groups), the tear volume, TBUT, corneal fluorescein staining at 7 and 14 days, conjunctival goblet cell density, and corneal apoptotic cell counts at 14 days improved in all treatment groups compared to EDE, with a significantly better result in the CsA-CL group compared with other groups (all *p* < 0.05). The CsA-CL could be an effective, stable, and safe option for inflammatory dry eye.

## 1. Introduction

Dry eye (DE) is characterized by the loss of homeostasis of the tear film, tear hyperosmolarity, the inflammation of the ocular surface, and epithelial damage accompanied by symptoms such as discomfort or visual disturbances [1,2,3]. Numerous factors, either extrinsic (e.g., a dry environment, prolonged exposure of the ocular surface, and refractive and cataract surgery) or intrinsic (e.g., aging, female sex, autoimmune disease, and systemic anticholinergics), can affect the vicious cycle of inflammation in DE [4,5]. Topical cyclosporine A (CsA) has proven its effectiveness in controlling inflammation, especially in treating moderate to severe DE refractory with artificial tears, improving both the symptoms and signs of DE [6,7,8,9,10,11]. However, topical CsA has potential side effects such as ocular pain, a burning sensation, and hyperemia, making it challenging for patients to apply eye drops on schedule [6,12]. Moreover, the drug penetration of CsA is typically less than 5% due to its poor water solubility and large molecular size [13,14].

Applying a therapeutic contact lens (CL) is also an option for the treatment of DE [15,16]. In addition to its protective properties, attempts have been made to embed the drug into the CL in order to deliver it at a higher concentration than topical eye drops and to reduce the need to apply eye drops such as antibiotics, steroids, anti-VEGF, anti-glaucoma medication, and CsA as frequently [17,18,19,20,21,22]. Recently, a drug-eluting contact lens with ketotifen was FDA-approved for the first time in the US, Japan, and Canada and is commercially available [23]. However, utilizing CLs to deliver drugs is still challenging due to their high water content. This unique trait leads to short drug release periods of only a few hours, which is shorter than the time required for therapeutic application [24,25,26]. Furthermore, when carried in a storage solution to prevent brittleness, the drug which is loaded on the CL has the potential of being released, making it difficult to predict the exact dose delivered to the patient [24,25,26].

Cellulose acetate phthalate (CAP) is a derivative of cellulose ester. Due to its dissolving nature in aqueous media of pH 6 and above, it has been used as an enteric coating material for tablets and capsules to delay drug release at the gastric track [27]. The distinct properties of CAP make it capable of loading hydrophobic drugs, protecting them from the environment and permitting the release of the loaded drugs in slightly neutral to alkaline physiological conditions [28]. By fabricating a pH-sensitive microporous matrix structure, CAP can be used as a suitable excipient for the development of a novel drug delivery system with long residence times and specific localization [28,29,30,31,32].

Previously, our team developed a CsA-eluting CL with a nanoporous silica carrier using a supercritical fluid technique and demonstrated its effectiveness in a rabbit DE model [22]. In this study, we advanced that technique by inventing a pH-responsive CsA-loaded CL that continuously releases CsA without an initial burst when exposed to a physiological pH environment, and does not release CsA in a storage environment. We aimed to evaluate the efficiency, safety, and stability of the pH-responsive CsA-loaded CL in the rabbit DE model.

## 2. Results

### 2.1. Properties of pH-Responsive CsA-Loaded CL

CsA was loaded on the pH-responsive CL by printing the CsA-CAP pH-responsive matrix solution on the CL. By changing the number of printings, different amounts of CsA were loaded onto the CL. The amount of CsA on the CL was adjustable from 20 to 115 μg. For the release test, we chose the CL with the highest CsA loaded. The amount of CsA on each pH-responsive CL was 114.23 ± 4.18 μg. The in vitro release of CsA from the CL was measured with HPLC and is presented in Figure 1. The concentration of CsA in the storage media was measured under a storage environment, at pH 5.4 and 4 °C. The concentration of CsA in the storage solution was 0 (not detectable) until 24 h and it was approximately 4.29% of the initial drug load after 90 days in the storage environment. When a pH-responsive CsA-loaded CL (CsA-CL) was exposed to a physiological mimicking environment (pH 7.4, 37 °C), 87.37% of the initially loaded CsA was continuously released for 12 h without an early burst of CsA, followed by a slow release of CsA for the remaining 12 h, reaching 94.03% at 24 h.

Safety profiles of the CsA-CL were evaluated. In the cytotoxicity assay, the viability of cultivated cells in an MTT assay were 98.93 ± 2.74%, 95.57 ± 2.80%, 96.40 ± 2.66%, 95.31 ± 2.25%, and 92.60 ± 2.69% in concentrations of 0.01%, 0.05%, 0.10%, 0.50%, and 1.00%, respectively. Cytotoxicity was not notable at concentrations of 1.00% or less. In the bovine corneal opacity and permeability (BCOP) assay for ocular irritancy, changes in corneal opacity or corneal permeability were not notable during the experiment. IVIS was −0.2 ± 0.5 in the testing sample, −0.2 ± 0.4 in the negative control, and 39.5 ± 2.5 in the positive control confirming that the sample was safe from ocular irritation. The water content of the CL was 38%. The average visible light transmittance of the CsA-CL was 93%, which was over the ISO guideline minimum (80%). The average oxygen permeability was 49.02 Dk.

### 2.2. In Vivo Release of CsA

CsA concentrations in the cornea, conjunctiva, and crystalline lens at 2, 6, and 12 h are described in Figure 2. The concentration in the tissues increased over time. The concentrations were significantly higher in the conjunctiva than the cornea and the crystalline lens at 6 and 12 h (*p* < 0.05). Sufficient uptake by the crystalline lens proved that there was good penetration of the drug into the tissues.

### 2.3. Tear Film and Ocular Surface Parameters

The tear volume did not significantly differ between groups except for the normal control (NC) group at baseline. The mean tear volume at 7 and 14 days significantly decreased in the experimental dry eye (EDE) group (4.38 ± 0.41 mm and 5.33 ± 1.21 mm) compared with the NC group (23.83 ± 2.40 mm and 24.83 ± 1.94 mm; *p* < 0.01). The mean Schirmer`s test values of the treatment groups at 7 and 14 days were 12.83 ± 2.04 and 14.33 ± 1.75 mm in the CsA eye drop (CsA-ED) group, 11.00 ± 1.79 and 13.50 ± 2.17 mm in the naïve contact lens (CL) group, and 17.83 ± 1.72 and 20.33 ± 1.86 mm in the CsA-CL group, respectively (Figure 3). The CsA-ED, CL, and CsA-CL groups showed significantly higher tear volume values compared to the EDE group at 7 and 14 days (all *p* < 0.05). The value was higher in the CsA-CL group compared to CsA-ED and CL groups at 7 and 14 days (all *p* < 0.05) (Figure 3). The tear volume significantly improved in the CsA-CL group at days 7 and 14 compared to baseline (both *p* < 0.01). At day 14, all treatment groups showed a significantly improved tear volume (all *p* < 0.05).

There were no statistically significant differences in the tear film break-up time (TBUT) among the groups except for the NC group at baseline. The mean TBUT at 7 and 14 days significantly decreased in the EDE group (4.38 ± 0.41 and 3.79 ± 0.48 s) compared to the NC group (8.49 ± 0.84 mm and 8.53 ± 0.93 mm; *p* < 0.01). The mean TBUT of the treatment groups at 7 and 14 days were 5.54 ± 0.44 and 6.26 ± 0.51 s in the CsA-ED group, 5.38 ± 0.44 and 6.02 ± 0.45 s in the CL group, and 6.85 ± 0.48 and 7.33 ± 0.54 s in the CsA-CL group, respectively (Figure 4). The CsA-ED, CL, and CsA-CL groups showed significantly longer TBUTs compared to the EDE group at 7 and 14 days (all *p* < 0.05). The values were higher in the CsA-CL group compared to the CsA-ED and CL group at 7 and 14 days (all *p* < 0.05). TBUT significantly improved at day 7 and 14 compared to the baseline in all treatment groups (all *p* < 0.05).

The corneal fluorescein staining scores were 1.50 ± 1.52, 2.33 ± 1.21, and 2.83 ± 1.94 in the NC group, 9.67 ± 1.03, 10.33 ± 1.51, and 11.33 ± 1.36 in the EDE group, 9.83 ± 0.75, 8.00 ± 1.41, and 7.17 ± 1.17 in the CsA-ED group, 9.83 ± 0.98, 8.00 ± 0.89, and 7.67 ± 1.51 in the CL group, and 9.83 ± 1.17, 5.83 ± 0.75, and 3.33 ± 0.52 in the CsA-CL group at baseline, 7 days, and 14 days, respectively (Figure 5). After treatment with CsA-ED, CL, and CsA-CL, there was a significant decrease in corneal staining scores compared to the EDE group (all *p* < 0.05). Moreover, the CsA-CL group showed a more significant reduction in the corneal staining score than other treatment groups at 7 and 14 days (all *p* < 0.05). At day 7, the values significantly improved in the CL and CsA-CL groups, and at day 14, the values improved in all the groups (all *p* < 0.05).

### 2.4. Histologic Analysis

Representative images of periodic acid–Schiff (PAS)-stained conjunctival tissues at 14 days in all the groups are presented in Figure 6A. The densities of conjunctival goblet cells were 150.02 ± 11.19, 12.29 ± 9.87, 80.17 ± 10.01, 39.31 ± 8.88, and 120.11 ± 11.37 cells/100 μm^2^ in the NC, EDE, CsA-ED, CL, and CsA-CL groups, respectively. The density was significantly higher in the treatment groups compared to the EDE group (all *p* < 0.05). The CsA-ED and CsA-CL group showed a significantly higher conjunctival cell density compared to the CL group, and the value in the CsA-CL group was higher than the CsA-ED group (all *p* < 0.05) (Figure 6B).

Representative images of corneal tissues at 14 days stained with TUNEL (green) and counterstained with DAPI (blue) are presented in Figure 7A. The corneal apoptotic cell counts were 1.16 ± 1.51, 22.33 ± 2.11, 11.43 ± 1.29, 13.98 ± 1.10, and 7.03 ± 1.41 in the NC, EDE, CsA-ED, CL, and CsA-CL groups, respectively. The cell count was significantly lower in the treatment groups compared to the EDE group (*p* < 0.05). The CsA-ED and CsA-CL groups showed significantly lower corneal apoptotic cell counts compared to the CL group, and the value in the CsA-CL group was lower than the CsA-ED group (all *p* < 0.05) (Figure 7B).

## 3. Discussion

The inflammatory etiology of DE has been implicated in many studies [2,3,4]. In the Tear Film and Ocular Surface Society’s Dry Eye Workshop II (TFOS DEWS II) report [15], a step-wise approach according to disease severity is recommended for the treatment of dry eye. After sufficient ocular lubrication and management of lid hygiene as a first step, topical medications such as topical steroids, topical secretagogues, and topical CsA are considered as a second step. Then, therapeutic CLs or soft bandage lenses are considered as a third step [15]. CsA is beneficial for improving tear production, reducing inflammatory markers, reducing elevated tear osmolarity, reducing apoptosis of epithelial cells, and recovering conjunctival goblet cell density in subjects with DE [6,33,34,35]. However, CsA eye drops have drawbacks such as low bioavailability due to CsA’s hydrophobic nature and large molecular size, resulting in less than 5% penetration to the ocular tissues [13,14]. Other barriers to sufficient drug delivery include physiological barriers such as blinking and dilution by the tears, poor adherence to treatment, and wasting eye drops [36,37]. To overcome these obstacles, various strategies have been used, including varied drug concentrations, frequent instillation, and different drug delivery technologies such as polymeric gels, liposomes, and nanoparticles [38]. Despite these efforts, compliance has been one of the major issues associated with CsA instillation. In this study, we attempted to overcome these obstacles by fabricating a pH-responsive CL-based drug delivery system.

Many studies attempted to embed drugs into the CL in order to deliver the drug at a higher concentration. Initially, and most commonly, soaking methods were adopted for antiglaucoma medication, antibiotics, antifungals, NSAIDs, and steroids; however, most of the drugs were only eluted for a short duration of time (less than 4 h) [39,40,41,42]. To overcome the drawbacks, drug-loaded films were trialed [19,21,21,43]. Other approaches include using drug-loaded nanoparticles, liposomes, and micelles, molecular imprinting, and the supercritical carbon dioxide technique [39,44,45,46]. In our previous study [22], the nanoporous silica and supercritical fluid technique were adopted for the fabrication of CsA-loaded CL. Despite improvements in clinical parameters with the developed CL, the system was unable to solve problems such as the initial burst and the drug release completion within 6 h [24,25,26]. The burst release of drugs is a major obstacle to controlling drug release as it reduces the total duration of the drug. Additionally, extreme burst release may potentially lead to toxicity and thereby raise a safety issue [47].

In this study, we developed a drug-delivery system based on a pH-responsive CL by loading CsA into the CAP matrix structure. It was confirmed that CsA was continuously released for 24 h without an initial burst, as seen in Figure 1. CsA was loaded onto the CL by printing the CsA-CAP solution on the CL surface. The loading amount of CsA was determined by printing the combined solution on the lens as many times as desired. The pH-responsive property of this system made it possible to prevent drug leakage during transport or storage, a feature that is critical for commercial translation, showing a 90-day stability with less than a 5% leakage of the loaded CsA. In safety assays, the CsA-CL was safe from cytotoxicity and ocular irritation. Furthermore, the lens demonstrated good water content, visible light penetration, oxygen penetration, diameter, and a base curve which were comparable to commercial contact lenses.

To observe the efficacy of the novel CsA-CL, we adopted a rabbit EDE model induced with instillation of BAC eye drops [48]. This model induces dry eye by damaging the cornea and conjunctiva, and accordingly results in a decrease in the aqueous component of tear film [49]. When 0.1% BAC was applied in rabbit eyes twice a day, significant decreases in Schirmer scores, conjunctival goblet cell density, and MUC5AC, and an increase in corneal fluorescein staining were noted on days 7 and 14 [48]. We instilled BAC for 2 weeks prior to the experimental period and confirmed the induction of EDE. We chose CsA-ED and naïve CLs, which are widely accepted treatments for DE, as comparative groups to CsA-CL. Bandage CLs help stabilize the tear film and assist in the restoration of normal epithelial cell turnover, resulting in an improvement in DE symptoms and corneal epitheliopathy [50]. It is presumable that naïve CLs in this study may help to reduce desiccating stress and work as a barrier against the additional instillation of BAC.

The pH-responsive CsA-CL showed good clinical efficacy in controlling EDE. The tear volume, TBUT, and corneal staining improved in the CsA-CL group at day 7 and 14, and the results were significantly better compared to other groups. The conjunctival goblet cell density was significantly higher than in other treatment groups in the CsA-CL group at day 14 and the TUNEL assay showed a lower number of apoptotic cells. Because we have overcome the initial burst release problem in the previous model [22], the concentration of CsA in the ocular tissues increased over time until 12 h after applying the CL. The findings suggest the following two possibilities: the CsA-CL delivered a sufficient amount of CsA into the ocular tissue, which was not achieved with instilling eye drops, and the additive effect of CsA and the therapeutic CL had a synergistic effect in treating EDE.

The concentration of CsA in the crystalline lens was significantly higher than that in the cornea and conjunctiva. The passive diffusion of topically administered drugs into the eye is hampered by lacrimal drainage, thermodynamic activity, corneal and conjunctival epithelial barriers, and drug extraction into the conjunctival and choroidal vessels [51]. Del Amo [52] also described the fate of eye drops as: drug clearance from the ocular surface, drug absorption into the inner tissues of the eye, and drug distribution within the eye and into systemic circulation. Drug distribution in tissues differs according to the drug lipophilicity and tissue properties [53]. As for CsA, Saha et al. [54] reported the existence of the p-glycoprotein drug efflux pump in rabbit conjunctival epithelial cells, which pumps CsA out of conjunctival epithelial cells. Additionally, the esterase activity in the albino rabbit cornea and iris was higher than that in the crystalline lens supporting the different distributions of the drug in the tissues [55]. The lens acts as a physical barrier to the vitreous from the anterior chamber and vice versa [56]. Furthermore, the lens can act as a drug reservoir, thereby affecting ocular pharmacokinetics [57]. Tang-Liu et al. [58] reported that some lipophilic drugs can penetrate to the lens body and show higher lens affinity. In this study, the CsA concentration in the cornea and conjunctiva accumulated over time, and that of the crystalline lens was stationary. One hypothesis is that CsA on the ocular surface, such as the cornea and conjunctiva, could be washed out continuously with blinking; however, CsA in the lens would not be affected. The CsA concentration continuously increased at the ocular surface because CsA was continuously released from the CsA-CL. Finally, CsA accumulated in the cornea and conjunctiva, reaching higher concentrations of CsA. However, the sample size was small and the result might be biased. We cannot preclude the occurrence of any mistakes during the tissue excision and extraction. We will need to conduct further studies with larger sample sizes.

Despite the promising findings of this study, there are some limitations. The concentration of CsA in the cornea, conjunctiva, and the lens after instilling CsA eyedrops was not measured. However, in a recent study conducted on rabbits [59], the highest concentration of CsA was less than 0.4 μg/mL in the cornea and conjunctiva from 0 to 72 h after instilling 0.05% CsA eye drops, which was less than that of the CsA-CL group in this study. Comparing a full day of wearing CsA-CLs with a twice-daily instillation of eye drops may not seem adequate. However, we aimed to compare the results in a clinical setting. The results of our study will provide more practical information. All procedures were performed under anesthesia and tear volume may have been underestimated. However, since this study compares between groups, this is not expected to affect the results significantly. In vivo eye irritation was not evaluated in this study and the length of the treatment was relatively short. Further studies on the long-term use of the CsA-CL and possible symptom changes associated with the use of CsA-CLs will provide a ladder for advancement into clinical research.

Conclusively, with further clinical studies, CAP-based pH-responsive CsA-loaded CLs could be an effective and safe solution to the poor compliance to CsA eye drops and the low bioavailability of the drug for dry eye patients.

## 4. Materials and Methods

### 4.1. Materials

Ethanol and methylene chloride were purchased from Fisher Scientific (Hampton, USA). CAP was obtained from Eastman chemicals (Kingsport TN, USA) and CsA was purchased from Taejoon Pharmaceutical Co. Ltd. (Seoul, South Korea). Hydroxyethyl methacrylate (HEMA), poly(ethylene glycol) methyl ether methacrylate(Mn 950), and ethylene glycol dimethacrylate (EGDMA) were obtained from Sigma-Aldrich (St Louis, MO, USA). Azobisisobutyronitrile (AIBN) was procured from Junsei (Tokyo, Japan). CL polypropylene molds consisting of top and bottom molds with the diameter of 11.4 mm and the thickness of 0.1 mm were supplied by Youngwon Tech (Daegu, Korea).

### 4.2. Preparation of the CsA-Loaded pH-Responsive CAP Matrix and CsA-CL

In order to prepare the CsA-loaded pH-responsive CAP matrix solution, the dissolving media were prepared by combining methylene chloride (30%, *v/v*) and ethanol (70%, *v/v*) in a glass container. The measured amounts of CAP (150 mg/mL) and CsA (125 mg/mL) were added to the dissolving media, and were then allowed to settle for 10 min to moisten the CAP and CsA in the media. The mixture was then agitated for 20 min at room temperature on a magnetic stirrer to produce a CsA-loaded pH-responsive CAP matrix solution for the pH-responsive CsA-CL.

We prepared the CL by modifying the method described by Choi et al. [22]. The monomers were initially vacuum-distilled before polymerization. Subsequently, the CL coating solution was prepared by mixing poly(ethylene glycol) methyl ether methacrylate (5 g), HEMA (4.92 g), EGDMA (0.04 g), and AIBN (0.04 g). The CL solution was prepared by mixing HEMA (9.92 g), EGDMA (0.04 g), and AIBN (0.04 g). We then placed the mixture into a vacuum to remove air bubbles. The CL coating solution was printed on the top mold using a silicon pad and heated at 120 °C for 20 min to prepare the upper CL part. The coating solution containing the CsA-loaded pH-responsive CAP matrix solution was additionally printed on the prepared upper CL part using a silicon pad and heated at 120 °C for 20 min. Following the injection of the CL solution into the bottom mold, it was combined with the top mold and heated at 120 °C for 20 min to allow polymerization to occur. After cooling to room temperature, samples were removed from the molds and placed in 400 mL of deionized water. To completely remove unreacted monomers and initiators, the washing step was carried out for 2 days by changing the water three times per day. Finally, the pH-responsive CsA- CL was soaked in PBS and sterilized at 121 °C for 30 min (Figure 8B). The mold used for the preparation of CLs had a 14.2 mm outer diameter with an 8.6 mm base curve. The Chiltern Lens Analyzer (Optimece JCF, Optimece Ltd., Malvern, United Kingdom) was used to measure the diameter and base curve of the prepared CL [60].

### 4.3. Characterization of pH-Responsive CsA-CL

A modified method was used for the characterization of the CsA-CL [61]. The loading concentration of CsA on the CL was evaluated with the modified extraction method. Briefly, CsA-CLs were placed deep into 10 mL of the solution composed with methylene chloride (30%, *v/v*) and ethanol (70%, *v/v*), followed by water bath sonication at 4 °C, for 30 min. In total, 2 mL of the solution was taken and filtered through a 0.45 µm syringe filter. The filtered solution was injected into the HPLC system and the CsA concentration was analyzed as presented in ‘Quantitative analysis of CsA’ [61].

The modified United States Pharmacopeia (USP) dissolution method, dissolution apparatus 1 (rotating basket), was used to assess the drug release characteristics of the developed CL in phosphate buffer saline (PBS) under various release conditions at 37 °C, pH 7.4, simulating the physiological environment, and at 4 °C, Ph 5.4 for the CL storage condition. For the sink condition, one percent (*w/v*) of sodium lauryl sulfate (SLS) and 0.02% of Na-azide were added and combined as preservatives. A total of 200 mL of the release media filled the dissolution vessel. CLs were placed within the basket, which was then deep inside the vessel and rotated at a speed of 50 rpm. The sampling was performed at pre-determined time points of 0, 0.5, 1, 3, 6, 12, and 24 h. Each time, 1 mL of the sample was taken and 1 mL of the fresh release media was added into the vessel. The collected samples were filtered the through 0.45 µm syringe filter and then analyzed as presented in ‘Quantitative analysis of CsA’. Additional analysis of the CsA concentration in the media after 90 days of storage at 4°C, pH 5.4 was completed [62,63].

CsA was quantitatively analyzed with the Azura HPLC system (Berlin, Germany). The HPLC system consisted of an Azura pump P 6.1 L, an Azura autosampler AS 6.1 L, an Azura detector DAD 6.1 L, an Azura column thermostat CT 2.1, and a Luna^®^ 5μm C18 100 Å, 250 × 4.6 mm column (Phenomenex, Torrance, USA). The column was placed in the column oven set at 50°C and the UV detection wavelength was set at 210 nm for CsA. The sample injection volume was 20 µL with a run time of 20 min. Gradient flow was used to analyze the samples. The mobile phase was a mixture of acetonitrile and methanol, and the flow rate was 1 mL/min. Solvent A consisted of a mixture of acetonitrile and methanol at a volume ratio of 62.5:37.5 with 0.5% H_3_PO_4_, and solvent B was consisted with 0.5% H3PO4. The following gradient flows were used: from the start to 3 min 20% of A and 80% of B; from 5–15 min, 95% of A and 5% of B; and from 17–20 min: 20% of A and 80% of B [62].

To evaluate cytotoxicity, the MTT assay, according to the ISO 10993-5:2009(E) guidelines, was performed with Statens seruminstitut rabbit cornea in an authorized testing institute (Korea Testing & Research Institute; KTR, Hwasun, Korea). The samples for cytotoxicity testing were prepared by diluting the soaking solution from eluting the CsA-loaded CL in PBS for 24 h at 37 °C. A cell viability of less than 70% after cultivating the cells at concentrations of 0.01%, 0.05%, 0.10%, and 1.00% of the CsA-CL extract was considered toxic [64].

To evaluate in vitro ocular irritancy, a bovine corneal opacity and permeability (BCOP) assay was performed to evaluate the ocular irritancy of CsA-CLs in KTR. Freshly extracted bovine corneas were placed on a cornea holder and a 750 μL solution of water (negative control), CsA-CL extract (testing sample), and ethanol (positive control) was placed on the corneas. After 10 min, the corneas were irrigated and the opacity and permeability were evaluated with 490 nm wavelength light with a multi-channel microplate spectrophotometer (Epoch, Bio Tek, VT, USA) to calculate the in vitro irritancy score (IVIS) [65].

Visible light transmittance was measured with a spectrometer (OPTIZEN POP, Klab inc., Daejeon, Korea). The CsA-CL was mounted in the quartz cuvette filled with PBS at 20 °C ± 0.5 °C for 24 h and the relative light transmittance (%) to an empty cuvette was measured [22].

The oxygen permeability of CsA-CLs was measured according to the International Organization Standardization (ISO) guidelines [66]. After being placed in PBS at 20 °C ± 0.5 °C for 24 h, the samples were additionally placed at 35 °C ± 0.5 °C for another 2 h. The polarographic method was used in three sets of tests on 4 selected lenses of different thicknesses and the permeability was presented in Dk [10^−11^ (cm^2^/s) mL O2/(mL·mmHg)].

The water content of the developed contact lenses was determined by weighing the contact lenses in dry and wet conditions [67]. At room temperature, the completely dry CL weight was taken. Then, the CL was soaked in phosphate-buffered saline (PBS, pH 7.4) for 24 h so that it became swollen completely. The CL was taken from the PBS and blotted with tissue paper to remove surface water. Then, the weight of the swollen CL was taken and water content was calculated using the following formula:(1)$$\mathrm{Water}\;\mathrm{content}\;\left(\%\right)=\frac{\mathrm{Ws}\;-\;\mathrm{Wd}}{\mathrm{Wd}}\times 100\;$$
where W_s_ and W_d_ represent CL weights in the swelling and dry conditions, respectively. The experiment was repeated three times to obtain the average value for each sample. Generally, the CL contains 30–80% water depending on its construction materials [68].

### 4.4. Preparation of Animal Model

As described in our previous study [22], fully awake female New Zealand albino rabbits (Orient Bio Inc., Seongnam, South Korea) were prepared in standard cages at 19 °C ± 1 °C and 50% ± 5% relative humidity, with proper light conditions (12-hour light–dark cycle). The study was approved by the Committee on Care and Use of Laboratory Animals at Chonnam National University, Gwangju, South Korea (CNUHIACUC-20022). The rabbits were randomly assigned to one of the following 5 groups: the normal control (NC) group, experimental dry eye (EDE) group, 0.05% CsA eye drop-treated (CsA-ED) group, CL group, and CsA-loaded pH-responsive CL (CsA-CL) group. EDE was induced in 4 groups except for the NC group before treatment by instilling 0.1% benzalkonium chloride (BAC; Sigma-Aldrich, St. Louis, MO, US) solution, twice daily in both eyes for 2 weeks [48]. After the induction of EDE, BAC was continuously administered twice daily during 2 weeks of the experimental period. In the CsA-ED group, BAC was administered after 1 h of CsA-ED administration and in the CL and CsA-CL groups, BAC was administered on the CL-applied eyes. Additionally, 6 more EDE rabbits were separately prepared for in vivo drug release measurements.

### 4.5. Measurement of In Vivo CsA Release in the Cornea, Conjunctiva, and Crystalline Lens

Separately prepared EDE rabbits were anesthetized and placed in a conventional holder and CsA-CLs were applied in both eyes. The rabbits were taken to obtain cornea, conjunctiva, and crystalline lens samples at 2, 6, and 12 h after applying the CsA-CLs. Two rabbits at each time were sacrificed by injecting intravenous thiopental sodium with the consequential enucleation of both eyes. The cornea, perilimbal conjunctiva, and crystalline lenses of the rabbits were harvested under a surgical microscope (Nikon Instruments Inc., Tokyo, Japan) and under aseptic conditions. The tissues were soaked in methanol after cutting them in small pieces with scissors and they were frozen immediately. The concentration of CsA was analyzed using HPLC [69].

### 4.6. Experimental Procedures in the Rabbit Dry Eye Model

All procedures, including eye drop administration, applying and removing CLs, and evaluation of tear volume, TBUT, and corneal fluorescein staining were performed under anesthesia with intramuscular ketamine hydrochloride (30 mg/kg) (Yuhan, Seoul, South Korea) and xylazine (6 mg/kg) (Bayer, Leverkusen, Germany) injection [22]. After 2 weeks of the induction of EDE, the CsA-ED group was treated with 50 µL of 0.05% CsA eye drops (Restasis^®^, Allergan Inc, Irvine, CA, USA), twice daily. The CL group was treated with naïve soft silicone hydrogel CLs, fabricated with the same mixtures used in fabricating CsA-CLs, only without printing the CsA-loaded CAP matrix solution. They were treated in both eyes, by simply applying the CL on the surface of the rabbit eye and leaving it for 24 h, and replacing the lens daily. The CsA-CL group was treated with a daily exchange of CsA-CLs for 14 days. After 7 and 14 days, clinical parameters including the tear volume, tear film break-up time (TBUT), and corneal fluorescein staining scores were measured. At 14 days after treatment, all rabbits were euthanized for histologic analysis.

### 4.7. Evaluation of Tear Volume, Tear Film Break-Up Time, and Corneal Fluorescein Staining

Information on clinical parameters was obtained in all groups after anesthetizing the rabbits according to the methodology described in our previous study [22]. Tear volume was evaluated with sterile Schirmer’s test strips (ColorBar™; Eagle Vision Inc., Memphis, TN, US) placed in the lower fornix for 5 min [70,71]. After instilling 5 µL of 1% sodium fluorescein in the inferior conjunctival sac using a micropipette, TBUT was recorded in seconds under cobalt blue light in slit lamp biomicroscopy (BQ-900; Haag-Streit, Bern, Switzerland) [72]. After ninety seconds, punctate staining of the corneal epithelium was evaluated and scored in the range of 0 to 16 points [22,73].

### 4.8. Histologic Analysis

Approximately 5 × 5 mm sized biopsy specimens were obtained from the superotemporal side of the bulbar conjunctiva, near the limbus, in both eyes of all the groups. Conjunctival tissues were fixed in formalin solution, embedded in paraffin, and cut into 4 µm thick sections at room temperature. The sections were processed and mounted on gelatin-coated slides after de-paraffinization and were then stained with periodic acid–Schiff (PAS) [74,75]. The numbers of conjunctival epithelial goblet cells were counted with a bright-field microscope (BC-51; Olympus, Tokyo, Japan) with a magnification of 40× along the length of three separate tissues by two independent masked observers [74]. A TUNEL assay for detecting DNA fragmentation found in the apoptotic cascade was performed in corneal tissue preparations embedded in paraffin. A commercially available kit (DeadEnd Fluorometric TUNEL System; Promega, Madison, WI, US) was used according to the manufacturer’s protocols with modifications [76,77]. Representative images were captured with a Leica laser scanning confocal microscope (Leica Microsystems, Heidelberg, Germany). TUNEL-positive cells in corneal epithelium were counted and the results were expressed as the average of TUNEL-positive cells from three sections per eye.

### 4.9. Statistical Analysis

All statistical analyses were performed using the Statistical Package for the Social Sciences 18.0 (SPSS Inc., Chicago, IL, US). A Kolmogorov–Smirnov test was used for verification of normal distribution. A repeated measure analysis of variance (RM-ANOVA) and Dunnett’s post hoc test were used to analyze the differences between groups. A paired T-test was performed to analyze the change in Tear volume, TBUT, and corneal fluorescein staining from baseline and at day 7 and 14. The level of significance was set at *p* < 0.05. All results are expressed as the mean ± standard deviation.

## Figures and Tables

**Figure 1 ijms-24-02361-f001:**
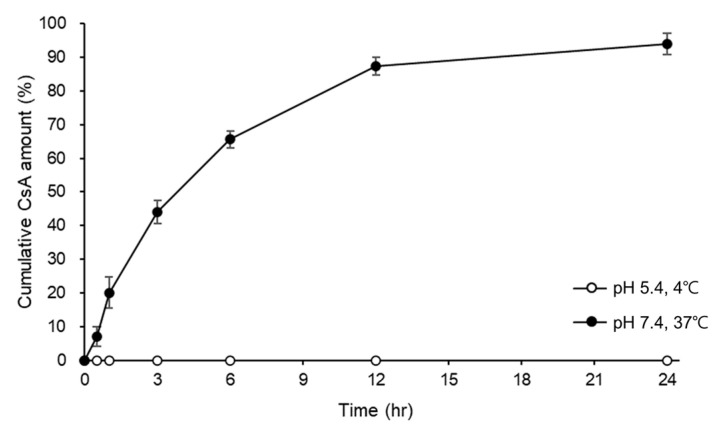
Cyclosporine A (CsA) release profile for pH-responsive CsA-loaded contact lenses in pH 5.4 at 4 °C or in pH 7.4 at 37 °C. All data are presented as mean ± SD.

**Figure 2 ijms-24-02361-f002:**
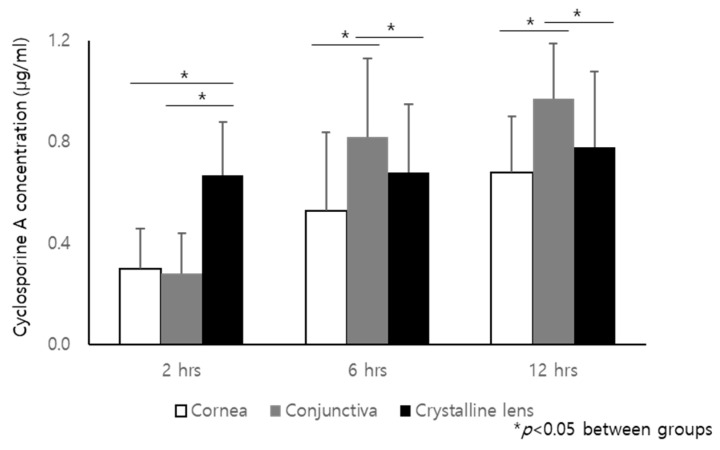
The concentration of cyclosporine A (CsA) in the cornea, conjunctiva, and crystalline lens of the rabbits at 2, 6, and 12 h after applying the CsA-loaded pH-responsive contact lens. All data are presented as mean and SD.

**Figure 3 ijms-24-02361-f003:**
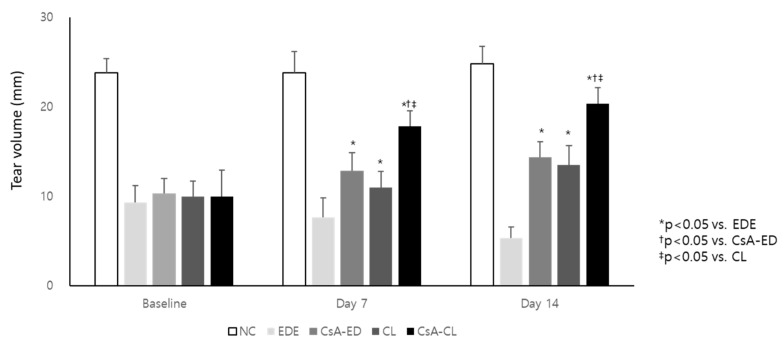
The tear volume at baseline and after 7 and 14 days of treatment. All data are presented as mean and SD. NC: normal control; EDE: experimental dry eye; CsA-ED: cyclosporine A eye drop; CL: contact lens; CsA-CL; cyclosporine A-loading pH-responsive contact lens.

**Figure 4 ijms-24-02361-f004:**
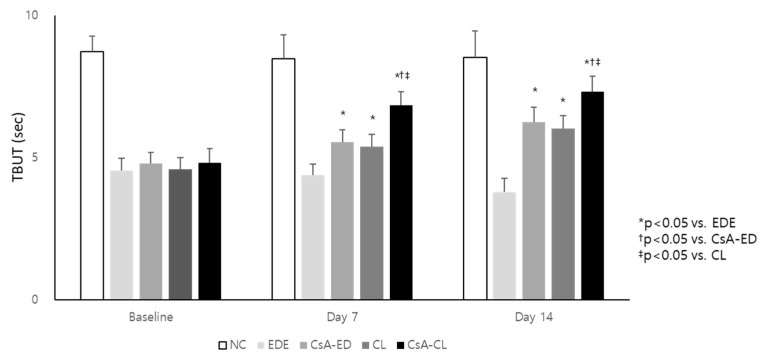
The tear film break-up time (TBUT) at baseline and after 7 and 14 days of treatment. All data are presented as mean and SD. NC: normal control; EDE: experimental dry eye; CsA-ED: cyclosporine A eye drop; CL: contact lens; CsA-CL: cyclosporine A-loading pH-responsive contact lens.

**Figure 5 ijms-24-02361-f005:**
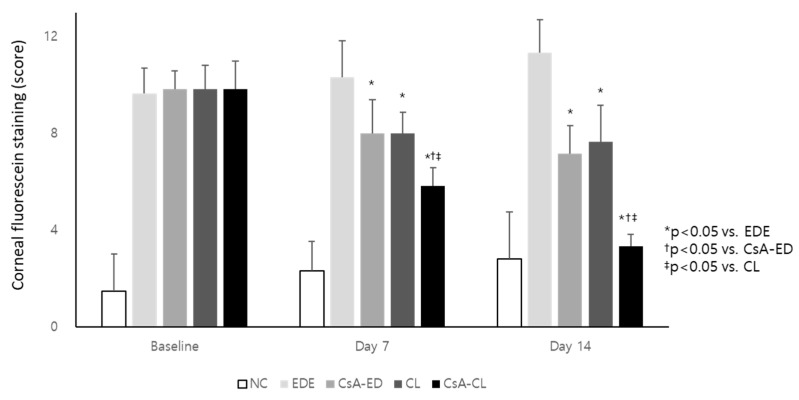
The corneal fluorescein staining at baseline and after 7 and 14 days of treatment. All data are presented as mean and SD. NC: normal control; EDE: experimental dry eye; CsA-ED: cyclosporine A eye drop; CL: contact lens; CsA-CL; cyclosporine A-loading pH-responsive contact lens.

**Figure 6 ijms-24-02361-f006:**
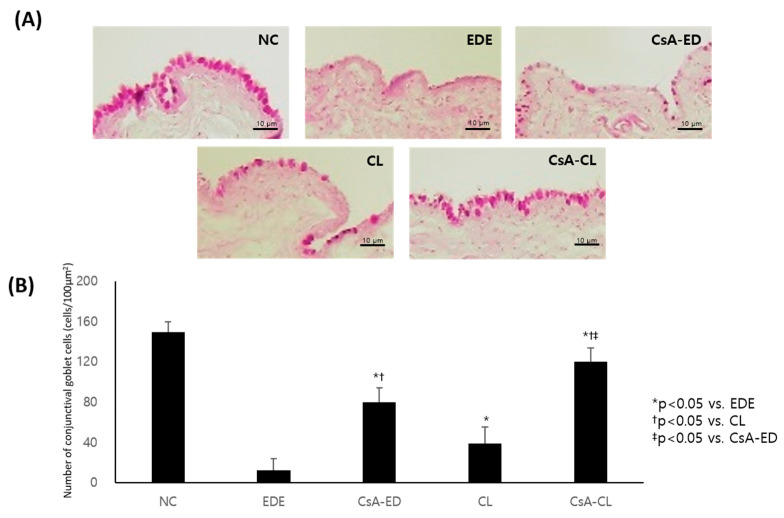
The representative images of PAS-stained conjunctival tissues obtained from all groups at 14 days (**A**) and the mean density and SD of conjunctival goblet cells in the samples presented in cells/100 μm^2^ (**B**). NC: normal control; EDE: experimental dry eye; CsA-ED: cyclosporine A eye drop; CL: contact lens; CsA-CL: cyclosporine A-loading pH-responsive contact lens.

**Figure 7 ijms-24-02361-f007:**
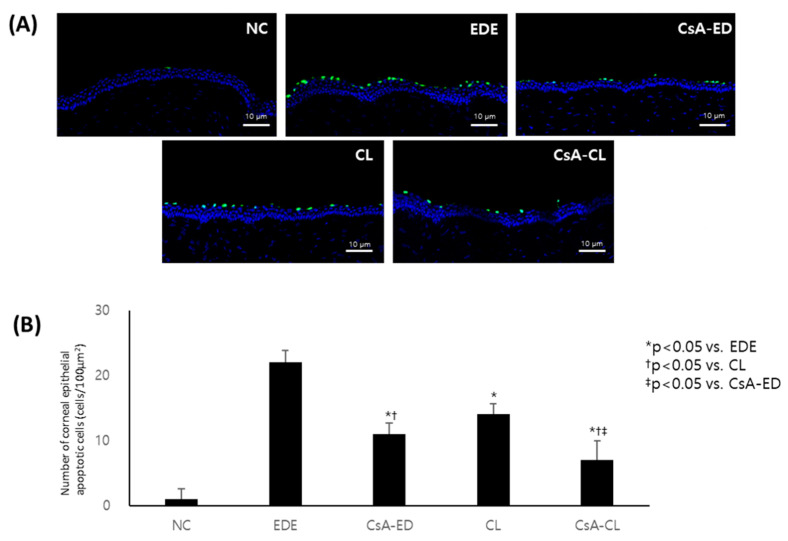
The representative images of corneal tissues at 14 days stained with TUNEL (green) and counterstained with DAPI (blue) (**A**) and the density of apoptotic cells in the samples (**B**). All data are presented as mean and SD. NC: normal control; EDE: experimental dry eye; CsA-ED: cyclosporine A eye drop; CL: contact lens; CsA-CL: cyclosporine A-loading pH-responsive contact lens.

**Figure 8 ijms-24-02361-f008:**
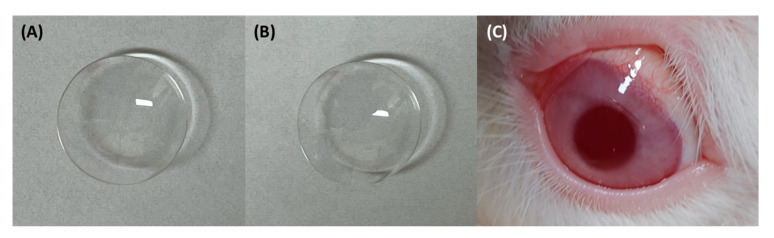
The picture of the naïve CL (**A**), CsA-CL (**B**), and CsA-CL-applied rabbit eye (**C**). CL: contact lens; CsA-CL: cyclosporine A-loading pH-responsive contact lens.

## Data Availability

Not applicable.

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
