# Peer review of "Cellulose Acetate Phthalate-Based pH-Responsive Cyclosporine A-Loaded Contact Lens for the Treatment of Dry Eye"

_ijms, 2023, doi:10.3390/ijms24032361_

Round 1

Reviewer 1 Report

This is a very interesting article 

Author Response

Thank you for your comment. 

Reviewer 2 Report

This manuscript demonstrated the short term efficacy and safety of CsA-CL in vivo and ex vivo. This manuscript rationalized the novel method in developing a slow and long-term release (up to 24 h) of CsA in order to provide a consistent concentration of CsA on the ocular surface for treating experimental dry eye. This paper is well structure but there are several points will need to be clarified. 

1.     It is unclear that what is naïve CL group, what were the materials of the CL? Why CL only was considered as a treatment group since CL can cause DE as well? It would be helpful the authors explained why CL only can improve tear volume/.

2.     What is the rationale of using BAK for 2 weeks as the experimental dry eye? It is likely caused dry eye due to the use of preservative? Why the environmental stressed was not used instead? Also, for the “treatment group” – suggesting calling them experimental groups? – Please clarify if the EDE animals were still given BAK or not on top of the CsA, CL and CsA-CL. If not, the findings of this study are confounded by not treating the treated EDE animals in the same situation as EDE animals – the recovery could be merely due to the cease of BAK use. 

3.     How did the authors determine the concentration used in the CLs? Also, for the cytotoxicity (check spelling on line 82), please provide the CsA concentration corresponding to the % use. Also, was it an eyedrop from or a CL form to deliver the drug for the assay?

4.     In Figure 2, it showed that the concentration of CsA was higher at 2 hours and remained the same level over the course of 12 hours. Can the authors provide an explanation for it? I would expect that the concentration would be higher in cornea and conj before the lens but apparently that is not true. Please provide references along with the authors’ speculation on the responses. 

5.     In terms of the CsA-CL, it would be helpful providing the real image of the CsA-CL so the readers can compare the designs with current available drug delivering contact lenses. It can be assumed that CL fitting was not considered in the study? Also, please explain if the contact lenses (both CL only and CsA CL) are dissolvable or the authors required to changes the lenses every 12 hours during the experiment? Were the animals anesthetized the whole time during the experiment or only during the CL insertion. How the authors kept the CL on the eye? If sutures were used, did other groups of the animals were also have sutures performed? 

Other comments

1.     Please provide proper references in the methods section. The authors missed almost all the methodologies citation in the methods.

2.     Please provide which side(s) of the conj (T/N/S/I) were used for histological analysis and the density of GC is in cells/mm2?

3.     Please provide the time point post-Tx for each figure legend in Figure 6 and 7 and in the histological section. 

4.     Abstract: not clear what is the treatment groups.

5.     Figure 1: Why there was no SD or error bar for 4C PH5.4? What are the error bars – SD? Please also clarify the definition of error bars in all figures. 

6.     Line 87: missing number of ±

7.     Missing baseline results in staining in text. 

8.     Please provide the P values of changes overtime from day 0, day 7 and day 14.

9.     Line 191: please provide the range of duration in drug release.

10.  Lines 198-200: different font

11.  Line 331: what is UT: untreated? Should it be EDE group?

Author Response

English editing is done.

Response to Reviewer 2 Comments

Thank you for your gracious review. We appreciate your comments and have addressed all the suggestions that were mentioned. These are the responses to the detailed comments given earlier.

Point 1: It is unclear that what is naïve CL group, what were the materials of the CL? Why CL only was considered as a treatment group since CL can cause DE as well? It would be helpful the authors explained why CL only can improve tear volume/.

Response 1: I appreciate your insightful comment. The mixture of materials described in the method section, lines 256-257 constitute a silicone hydrogel CL with sufficient oxygen permeability and with enhanced ability to load CAP-based drug delivery system. The naïve CL used in this study was fabricated in the same way with CsA-CL, the only difference being that the CsA-loaded CAP was not printed.

In the method section, there were parts of process that were not accurately explained in the original manuscript. All groups except the NC group were instilled BAC 2 times a day throughout the whole experimental period, either 1 hr after instilling CsA-ED for the CsA-ED group or on the CL for the CL and CsA-CL groups.

In TFOS DEWS II report1, bandage contact lens is a Level 3 therapeutic option for dry eye. It is suggested that silicone hydrogel contact lenses may help stabilize the tear film, and help restoration of normal epithelial cell turnover, resulting in improvement of dry eye symptoms and corneal epitheliopathy.2 It is presumable that naïve CLs in this study may help to reduce desiccating stress and work as a barrier against the additional instillation of BAC.

We explained the procedure in detail including continuous instillation of BAC in the study groups in the method section, lines 404-408 as follows:

“After the induction of EDE, BAC was continuously administered twice-daily during 2 weeks of the experimental period. In the CsA-ED group, BAC was administered after 1 hr of CsA-ED administration and in the CL and CsA-CL groups, BAC was administered on the CL-applied eyes.”

We clarified the property of naïve CL in the method section, lines 426-430 as follows:

“The CL group was treated with naïve soft silicone hydrogel CLs, fabricated with the same mixtures used in fabricating CsA-CL, only without printing the CsA-loaded CAP matrix solution, in both eyes, by simply applying the CL on the surface of the rabbit eye and leav-ing it for 24 hrs, and replacing the lens daily.”

We discussed why naïve CL group had increased tear volume in the discussion section, lines 224-235 as follows:

“ To observe the efficacy of the novel CsA-CL, we adopted a rabbit EDE model induced with instillation of BAC eye drops [48]. This model induces dry eye by damaging the cor-nea and conjunctiva, and accordingly results in a decrease in the aqueous component of tear film [49]. When 0.1% BAC was applied in rabbit eyes twice a day, significant decreas-es in Schirmer scores, conjunctival goblet cell density, and MUC5AC, and an increase in corneal fluorescein staining were noted on days 7 and 14 [48]. We instilled BAC for 2 weeks prior to the experimental period and confirmed the induction of EDE. We chose CsA-ED and naïve CL, which are widely accepted treatments for DE, as comparative groups to CsA-CL. Bandage CLs help stabilize tear film and assist in the restoration of normal epithelial cell turnover, resulting in an improvement in DE symptoms and corneal epitheliopathy.[50] It is presumable that naïve CLs in this study may help to reduce desic-cating stress and work as a barrier against the additional instillation of BAC.”

Point 2: What is the rationale of using BAK for 2 weeks as the experimental dry eye? It is likely caused dry eye due to the use of preservative? Why the environmental stressed was not used instead? Also, for the “treatment group” – suggesting calling them experimental groups? – Please clarify if the EDE animals were still given BAK or not on top of the CsA, CL and CsA-CL. If not, the findings of this study are confounded by not treating the treated EDE animals in the same situation as EDE animals – the recovery could be merely due to the cease of BAK use.

Response 2: Thank you for pointing out. We adopted the rabbit dry eye model suggested by Xiong et al..3 Treatment of rabbit eyes with 0.1% BAK (BAC as in the manuscript) twice a day significantly reduced Schirmer scores and increased corneal fluorescein staining at days 5, 7, and 14 compared to untreated controls. Reduction in goblet cell density and MUC5AC were also noted.

In our experiment, BAC was administered 2 times a day throughout the study period including 2 weeks of dry eye induction period and another 2 weeks of experimental period. There was no cessation of BAC. We made this clear and added references in the method section, lines 404-408 as mentioned in response 1. We also mentioned the rationale of using BAC to induce DE in the discussion section, lines 224-229 as mentioned above.

Point 3: How did the authors determine the concentration used in the CLs? Also, for the cytotoxicity (check spelling on line 82), please provide the CsA concentration corresponding to the % use. Also, was it an eyedrop from or a CL form to deliver the drug for the assay? 

Response 3: Thank you for making a good point. We designed the CsA-CL to load as much CsA as possible so that sufficient amount of CsA can be delivered to the tissues. As mentioned in the introduction section line 37-38, topical CsA is very ineffective in terms of bioavailabity. So, we aimed to overcome the limitations of conventional CsA eye drops.

We added information on determining the loading amount of CsA in CsA-CL in the results section, lines 70-71 as follows:

“The amount of CsA on the CL was adjustable from 20 to 115 μg. For the release test, we chose the CL with the highest CsA loaded.”

There is a limitation to this study design: we compared 2 times of CsA eye drops a day with a full-day wearing of CL with heavily loaded CsA. However, this study was designed to imitate a practical clinical field, so we decided to use the eye drops as instructed for general dry eye patients. We stated this as a limitation of this study in the discussion section, lines 275-277 as follows:

“Comparing a full-day wearing of CsA-CL with twice daily instillation of eye drop may not seem adequate. However, we aimed to compare the results in a clinical setting. The results of our study will provide more practical information.”

Cytotoxicity test was run by an authorized testing institute (Korea Testing & Research Institute; KTR, Hwasun, Korea). The test was performed according to ISO 10993-5:2009(E) guidelines (tests for in vitro cytotoxicity). For the test, CsA-CL was soaked in the PBS solution for 24 hrs at 37℃and this solution was diluted and used. The exact concentrations regarding testing % of the solution were not provided. However, the ISO guideline classifies cell viability more than 70% in 1.00% solution as non-toxic which was more than 90% in this study. We revised the cytotoxicity testing method by adding the sample information in the method section, lines 360-362 as follows:

“The samples for cytotoxicity testing were prepared by diluting the soaking solution from eluting the CsA-loaded CL in PBS for 24 hours at 37°C.”

Point 4: In Figure 2, it showed that the concentration of CsA was higher at 2 hours and remained the same level over the course of 12 hours. Can the authors provide an explanation for it? I would expect that the concentration would be higher in cornea and conj before the lens but apparently that is not true. Please provide references along with the authors’ speculation on the responses.

Response 4: Thank you for making a good point.

The concentration of CsA in the crystalline lens was significantly higher than that of cornea and conjunctiva. Passive diffusion of topically administered drugs into the eye is hampered by lacrimal drainage, thermodynamic activity, corneal and conjunctival epithelial barriers, and drug extraction into conjunctival and choroidal vessels.4 Del Amo5 also described the fate of eye drops as: drug clearance from the ocular surface, drug absorption into the inner tissues of the eye, and drug distribution within the eye and into systemic circulation. The drug distribution in the tissues differs according to the drug lipophilicity and tissue properties.6 As for CsA, Saha et al.7 reported the existence of p-glycoprotein drug efflux pump in rabbit conjunctival epithelial cells which pumps CsA out of conjunctival epithelial cells. Also, the esterase activity in albino rabbit cornea and iris was higher than that in the crystalline lens.8

The lens acts as a physical barrier to the vitreous from the anterior chamber and vice versa.9 Also, lens can act as a drug reservoir, affecting ocular pharmacokinetics.10 Tang-Liu et al.11 reported some lipophilic drugs can penetrate to the lens body and show higher lens affinity.

In this experiment, CsA concentrations in the corneal and conjunctiva accumulated as time passes, and that of the crystalline lens was stationary. One hypothesis is that CsA on the ocular surface such as cornea and conjunctiva could have been washed out continuously with blinking whereas CsA in the lens was not affected. The CsA concentration of ocular surface continuously increased because CsA was continuously released from the CsA-CL. Finally, CsA was acculumated in the cornea and conjunctiva, reaching higher concentrations of CsA.

However, the sample size was small and there can be a bias to the result. We cannot preclude any mistakes during the tissue excision and extraction. We will need further studies with larger sample sizes.

We’re afraid this explanation does not fully satisfy the readers and we are considering excluding the results regarding CsA concentrations in the crystalline lens. Your valuable comments on this will be appreciated.

In the discussion section, we explained our perspectives with references in lines 246-269 as follows:

“The concentration of CsA in the crystalline lens was significantly higher than that of cornea and conjunctiva. The passive diffusion of topically administered drugs into the eye is hampered by lacrimal drainage, thermodynamic activity, corneal and conjunctival epithelial barriers, and drug extraction into conjunctival and choroidal vessels [51]. Del Amo [52] also described the fate of eye drops as: drug clearance from the ocular surface, drug absorption into the inner tissues of the eye, and drug distribution within the eye and into systemic circulation. Drug distribution in tissues differs according to the drug lipophilicity and tissue properties [53]. As for CsA, Saha et al. [54] reported the existence of p-glycoprotein drug efflux pump in rabbit conjunctival epithelial cells, which pumps CsA out of conjunctival epithelial cells. Additionally, the esterase activity in albino rabbit cornea and iris was higher than that in the crystalline lens supporting the different distribution of drug in the tissues.[55] The lens acts as a physical barrier to the vitreous from the anterior chamber and vice versa.[56] Furthermore, the lens can act as a drug reservoir, thereby affecting ocular pharmacokinetics [57]. Tang-Liu et al. [58] reported that some lipophilic drugs can penetrate to the lens body and show higher lens affinity. In this study, the CsA concentration in the corneal and conjunctiva accumulated over time, and that of the crystalline lens was stationary. One hypothesis is that CsA on the ocular surface, such as the cornea and conjunctiva, could be washed out continuously with blinking; however, CsA in the lens would not be affected. The CsA concentration continuously increased at the ocular surface because CsA was continuously released from the CsA-CL. Finally, CsA was accumulated in the cornea and conjunctiva, reaching higher concentrations of CsA. However, the sample size was small and the result might be biased. We cannot preclude the occurrence of any mistakes during the tissue excision and extraction. We will need to conduct further studies with larger sample sizes.”

Point 5: In terms of the CsA-CL, it would be helpful providing the real image of the CsA-CL so the readers can compare the designs with current available drug delivering contact lenses. It can be assumed that CL fitting was not considered in the study? Also, please explain if the contact lenses (both CL only and CsA CL) are dissolvable or the authors required to changes the lenses every 12 hours during the experiment? Were the animals anesthetized the whole time during the experiment or only during the CL insertion. How the authors kept the CL on the eye? If sutures were used, did other groups of the animals were also have sutures performed?

Response 5: Thank you for the suggestion. We developed the CL following the conventional CL preparation method and we added reference information. We added the photos of naïve CL, CsA-CL, and CsA-CL-applied rabbit eye with a legend in the method section, line 321. As you can see in the photos, CsA-CL and naïve CL are not different in appearance under gross inspection. All of the characterized properties, including water content, oxygen permeability, and visible light transmittance of CsA-CL were comparable to those of commercial CL which does not dissolve.

The rabbits were anesthetized once every day to apply a new CL and to measure tear volume, TBUT, and corneal staining. The CLs were 11.4 mm in diameter and the base curve was 8.4 mm. The average diameter of rabbit cornea is 11.75 mm and the base curve is about 7.3 mm which is a little less than that of human (7.7-7,8 mm) according to the study by Bozkir et al..12 No other methods such as tarsorrhaphy or sutures were used. Every time the CLs were replaced, it was confirmed that the previous CL was in place, and the CLs were never lost during the experiment. We also added a picture of a rabbit eye applied with CsA-CL in figure 8C to help comprehension.

We revised the method section, lines 427-431 as follows:

“The CL group was treated with naïve soft silicone hydrogel CLs, fabricated with the same mixtures used in fabricating CsA-CL, only without printing the CsA-loaded CAP matrix solution, in both eyes, by simply applying the CL on the surface of the rabbit eye and leav-ing it for 24 hrs, and replacing the lens daily.”

Point 6: Please provide proper references in the methods section. The authors missed almost all the methodologies citation in the methods.

Response 6: We made all corrections as you recommended. All the method sections were now properly cited to the original methods with updated reference list.

Point 7: Please provide which side(s) of the conj (T/N/S/I) were used for histological analysis and the density of GC is in cells/mm2?

Response 7: The conjunctiva was obtained from superotemporal side and the information is added in the text in line 445-446 as follows:

“Approximately 5×5 mm sized biopsy specimens were obtained from the superotem-poral side of bulbar conjunctiva, near the limbus, in both eyes of all the groups.”

We corrected the unit into [cells/mm2] in the figure legends and the text.

Point 8: Please provide the time point post-Tx for each figure legend in Figure 6 and 7 and in the histological section.

Response 8: We stated the time of tissue preparation in the legends and in the text as follows:

“Figure 6. The representative images of PAS-stained conjunctival tissues obtained from all groups at 14 days (A) and the density mean number and standard deviation of conjunctival goblet cells in the samples presented in cells/100 μm2 (B).”

“Figure 7. The representative images of corneal tissues at 14 days stained with TUNEL (green) and counterstained with DAPI (blue) (A) and the density of apoptotic cells in the samples (B). “

“Representative images of corneal tissues at 14 days stained with TUNEL (green) and counterstained with DAPI (blue) are demonstrated in figure 7A.”

Point 9: Abstract: not clear what is the treatment groups.

Response 9: The “treatment groups” stands for the groups excluding the normal control group and EDE group. To clarify the meaning, we stated that the results regarding treatment groups are “compared to EDE” in the abstract.

Point 10: Figure 1: Why there was no SD or error bar for 4C PH5.4? What are the error bars – SD? Please also clarify the definition of error bars in all figures.

Response 10: In response to the reviewer's suggestion, we clarified the definition of all figures’ legends mentioning average and standard deviation values. The concentration of CsA in the PBS with pH 5.4 stored in 4℃was 0 (not detectable) during first 24 hrs when analyzed by HPLC. We stated this in the result section, line 73 as follows:

“The concentration of CsA in the storage solution was 0 (not detectable) until 24 hrs…”

Point 11: Line 87: missing number of ±

Response 11: The missing value was added in line 88 as follows: 39.5 ± 2.5. Thank you for pointing out

Point 12: Missing baseline results in staining in text.

Response 12: The baseline results for corneal fluorescein staining were added in the text lines 131-135 as follows:

The corneal fluorescein staining scores were 1.50 ± 1.52, 2.33 ± 1.21, and 2.83 ± 1.94 in the NC group, 9.67 ± 1.03, 10.33 ± 1.51, and 11.33 ± 1.36 in the EDE group, 9.83 ± 0.75, 8.00 ± 1.41, and 7.17 ± 1.17 in the CsA-ED group, 9.83 ± 0.98, 8.00 ± 0.89, and 7.67 ± 1.51 in the CL group, and 9.83 ± 1.17, 5.83 ± 0.75, and 3.33 ± 0.52 in the CsA-CL group at baseline, 7 days, and 14 days, respectively (Figure 5).”

Point 13: Please provide the P values of changes overtime from day 0, day 7 and day 14.

Response 13: We performed additional Paired T-test to see if the changes were significant in all groups. In all treatment groups (CsA-ED, CL, and CsA-CL), Tear volume, TBUT, and corneal fluorescein staining significantly improved at day 14 compared to baseline (all p<0.05). At day 7, tear volume significantly improved in the CsA-CL group only (p=0.007) which further increased at day 14 (p=0.018, day 7 vs 14). TBUT significantly improved at day 7 in all treatment groups (all p<0.05).  Corneal fluorescein staining significantly improved in the CL and CsA-CL groups at day 7 (both p<0.05).

We added statistical analysis in the method section, lines 464-466 as follows:

“A paired T-test was performed to analyze the change in Tear volume, TBUT, and corneal fluorescein staining from baseline and at day 7 and 14.”

We stated this result in the result section, lines 115-117, 131-132, and 145-147 as follows:

“The tear volume significantly improved in the CsA-CL group at day 7 and 14 compared to baseline (both p<0.01). At day 14, all treatment groups showed significantly improved tear volume (all p<0.05).”

“TBUT significantly improved at day 7 and 14 compared to baseline in all groups (all P<0.05).”

“At day 7, the values significantly improved in the CL and CsA-CL groups, and at day 14, in all groups (all P<0.05).”

Point 14: Line 191: please provide the range of duration in drug release.

Response 14: We provided the range (less than 4 hrs) in the text.

Point 15: Lines 198-200: different font

Response 15: Thank you for your observation. We have made correction and now all are in same font in the mentioned area.

Point 16: Line 331: what is UT: untreated? Should it be EDE group?

Response 16: It’s a typo for NC. We corrected it. Thank you for pointing out.

Reference

  1. Jones L, Downie LE, Korb D, et al. TFOS DEWS II Management and Therapy Report. Ocul Surf. 2017;15(3):575-628. doi:10.1016/j.jtos.2017.05.006
  2. Russo PA, Bouchard CS, Galasso JM. Extended-Wear Silicone Hydrogel Soft Contact Lenses in the Management of Moderate to Severe Dry Eye Signs and Symptoms Secondary to Graft-Versus-Host Disease. Eye Contact Lens. 2007;33(3):144. doi:10.1097/01.icl.0000244154.76214.2d
  3. Xiong C, Chen D, Liu J, et al. A Rabbit Dry Eye Model Induced by Topical Medication of a Preservative Benzalkonium Chloride. Invest Ophthalmol Vis Sci. 2008;49(5):1850-1856. doi:10.1167/iovs.07-0720
  4. Sripetch S, Loftsson T. Topical drug delivery to the posterior segment of the eye: Thermodynamic considerations. Int J Pharm. 2021;597:120332. doi:10.1016/j.ijpharm.2021.120332
  5. del Amo EM. Topical ophthalmic administration: Can a drug instilled onto the ocular surface exert an effect at the back of the eye? Front Drug Deliv. 2022;2. Accessed January 9, 2023. https://www.frontiersin.org/articles/10.3389/fddev.2022.954771
  6. Balla A, Auriola S, Grey AC, et al. Partitioning and Spatial Distribution of Drugs in Ocular Surface Tissues. Pharmaceutics. 2021;13(5):658. doi:10.3390/pharmaceutics13050658
  7. Saha P, Yang JJ, Lee VH. Existence of a p-glycoprotein drug efflux pump in cultured rabbit conjunctival epithelial cells. Invest Ophthalmol Vis Sci. 1998;39(7):1221-1226.
  8. Heikkinen EM, del Amo EM, Ranta VP, Urtti A, Vellonen KS, Ruponen M. Esterase activity in porcine and albino rabbit ocular tissues. Eur J Pharm Sci. 2018;123:106-110. doi:10.1016/j.ejps.2018.07.034
  9. Maurice DM, Mishima S. Ocular Pharmacokinetics. In: Sears ML, ed. Pharmacology of the Eye. Handbook of Experimental Pharmacology. Springer; 1984:19-116. doi:10.1007/978-3-642-69222-2_2
  10. Heikkinen EM, Auriola S, Ranta VP, et al. Distribution of Small Molecular Weight Drugs into the Porcine Lens: Studies on Imaging Mass Spectrometry, Partition Coefficients, and Implications in Ocular Pharmacokinetics. Mol Pharm. 2019;16(9):3968-3976. doi:10.1021/acs.molpharmaceut.9b00585
  11. Tang-Liu DD, Richman JB, Liu SS. Lenticular uptake and distribution of xenobiotics and amino acids. J Ocul Pharmacol. 1992;8(3):267-277. doi:10.1089/jop.1992.8.267
  12. Bozkir G, Bozkir M, Dogan H, Aycan K, Güler B. Measurements of axial length and radius of corneal curvature in the rabbit eye. Acta Med Okayama. 1997;51(1):9-11. doi:10.18926/AMO/30804

Reviewer 3 Report

This study described a pH sensitive CsA loaded contact lens, and evaluated the efficacy on rabbit dry eye model. The whole study design seems comprehensive, while there are still some details that need to be addressed.

1.    For the CsA-CL, how about the water content information? How about the morphology and the appearance of the CsA-CL?

2.    For the rabbit studies, how you place the CL on the cornea surface? Did  the rabbits need to be anesthetized? Secondly, how long the CL would need to be placed on the ocular surface?

3.    In figure 2, it is quite confusing that the “Lens” means CsA-CL or the lens tissue in rabbit eyes. Please clarify that and make changes in the manuscript accordingly if necessary.

4.    Figure 3, the tear volume measurement study, were the tear strips inserted in the conjunctiva area when rabbits were asleep? Or awake? Typically, under anesthesia conditions, the tears secretion could be inhibited which may influence the results. Please clarify it.

Author Response

English editing is done.

Point 1: For the CsA-CL, how about the water content information? How about the morphology and the appearance of the CsA-CL?

Response 1: -Thank you for your thoughtful suggestion. The water content in the CL was 38%. We added the water content result in the section ‘2.1. Properties of pH-responsive CsA-loaded CL’, and also add the water content evaluation method in the ‘4.3. Characterization of pH-responsive CsA-CL’. The morphology of the CL was concave shape with round circular opening as can be seen in naked eye and the image of the developed CL was presented at Figure 8(B).

Point 2: For the rabbit studies, how you place the CL on the cornea surface? Did the rabbits need to be anesthetized? Secondly, how long the CL would need to be placed on the ocular surface?

Response 2: The rabbits were anesthetized during all procedures including CL application, Schirmer test, TBUT measurement, and evaluation of corneal staining. We simply applied the CLs on the surface of the rabbit cornea and the CLs stayed well in place. The CLs were replaced with a new one everyday, resulting in a 24-hr of CL wearing. We stated and clarified the procedures in the method section, lines 422-425 and 427-431 as follows:

“All procedures including eye drop administration, applying and removing CLs, and eval-uation of tear volume, TBUT, and corneal fluorescein staining were performed under anes-thesia with intramuscular ketamine hydrochloride (30 mg/kg) (Yuhan, Seoul, South Korea) and xylazine (6 mg/kg) (Bayer, Leverkusen, Germany) injection.”

“CL group was treated with naive soft CLs in both eyes, by simply applying the CL on the surface of the rabbit eye and leaving it for 24 hrs, and replacingchanging the lens daily. CsA-CL group was treated with daily exchange of CsA-CL for 14 days.”

Point 3: In figure 2, it is quite confusing that the “Lens” means CsA-CL or the lens tissue in rabbit eyes. Please clarify that and make changes in the manuscript accordingly if necessary.

Response 3: Thank you for pointing out. We changed the term “Lens” into “Crystalline lens” in the figure, legend, and result and method section of the manuscript to avoid confusion. Also we changed “lens” meaning CsA-CL into “CsA-CL” in the method section, line 414.

Point 4: Figure 3, the tear volume measurement study, were the tear strips inserted in the conjunctiva area when rabbits were asleep? Or awake? Typically, under anesthesia conditions, the tears secretion could be inhibited which may influence the results. Please clarify it.

Response 4: Thank you for the suggestion. The tear volume was measured under anesthesia and we added the statement in the method section, lines 422-425 as mentioned in response 2.

As you mentioned, tear volume can be underestimated under anesthesia. However, rabbits in all groups were anesthetized when measuring tear volume and given the comparative nature of this study, anesthesia will have little influence of the significance of the results. We stated this as a limitation of this study in the discussion section, lines 277-280 as follows:

“All procedures were performed under ansethesia and tear volume may have been under-estimated. However, since this study compares between groups, it is not expected to affect the results significantly.”

Round 2

Reviewer 2 Report

Thanks for the details responses. There are several minor comments that I have: 

1. Please put (treatment groups:) before CsA eyedrops on line 19 in the abstract. 

Line 135: the groups mean treatment groups?

2. it would be helpful providing the diameter and BC for CL in the text

3. Still missing references for section 4.3 (the first 4 paragraphs)

Author Response

Response to Reviewer 2 Comments

Point 1: Please put (treatment groups:) before CsA eyedrops on line 19 in the abstract. Line 135: the groups mean treatment groups?

 Response 1: Thank you for pointing out. We put “treatment groups:” before CsA eye drop in the abstract, line 19.

The groups in line 145 in the result section means the treatment groups. We revised the sentence to avoid confusion as follows:

“TBUT significantly improved at day 7 and 14 compared to baseline in all treatment groups (all P<0.05).”

Point 2: It would be helpful providing the diameter and BC for CL in the text

Response 2: We appreciate your thoughtful insight. The diameter and the base curve of the lens is provided in the method section, lines 319-322 as follows:

“The mold used for the preparation of CL had 14.2 mm outer diameter with 8.6 mm base curve. Chiltern Lens Analyzer (Optimece JCF, Optimece Ltd., Malvern, United Kingdom) was used to measure the diameter and base curve of the prepared CL [60].”

Point 3: Still missing references for section 4.3 (the first 4 paragraphs)

Response 3: Thank you for your observations. We revised those missing references for section 4.3 (references 60, 62-64)

Thank you again for your thoughtful comments.